# Effect of Zr_2_Al_4_C_5_ Content on the Mechanical Properties and Oxidation Behavior of ZrB_2_-SiC-Zr_2_Al_4_C_5_ Ceramics

**DOI:** 10.3390/ma16124495

**Published:** 2023-06-20

**Authors:** Qilong Guo, Liang Hua, Hao Ying, Ronghao Liu, Mei Lin, Leilei Li, Jing Wang

**Affiliations:** 1School of Civil Engineering, Northwest Minzu University, Lanzhou 730124, China287142749@xbmu.edu.cn (L.L.); 13609327860@126.com (J.W.); 2School of Civil Engineering, Lanzhou University of Technology, Lanzhou 730050, China; lzlglinmei@126.com

**Keywords:** ZrB_2_-SiC composite ceramics, Zr_2_Al_4_C_5_ layered compound, in situ reactions, mechanical properties, oxidation resistance

## Abstract

ZrB_2_-SiC-Zr_2_Al_4_C_5_ multi-phase ceramics with uniform structure and high density were successfully prepared through the introduction of in situ synthesized Zr_2_Al_4_C_5_ into ZrB_2_-SiC ceramic via SPS at 1800 °C. A systematic analysis and discussion of the experimental results and proposed mechanisms were carried out to demonstrate the composition-dependent sintering properties, mechanical properties and oxidation behavior. The results showed that the in situ synthesized Zr_2_Al_4_C_5_ could be evenly distributed in the ZrB_2_-SiC ceramic matrix and inhibited the growth of ZrB_2_ grains, which played a positive role in the sintering densification of the composite ceramics. With increasing Zr_2_Al_4_C_5_ content, the Vickers hardness and Young’s modulus of composite ceramics gradually decreased. The fracture toughness showed a trend that first increased and then decreased, and was increased by about 30% compared with ZrB_2_-SiC ceramics. The major phases resulting from the oxidation of samples were ZrO_2_, ZrSiO_4_, aluminosilicate and SiO_2_ glass. With increasing Zr_2_Al_4_C_5_ content, the oxidative weight showed a trend that first increased then decreased; the composite ceramic with 30 vol.% Zr_2_Al_4_C_5_ showed the smallest oxidative weight gain. We believe that the presence of Zr_2_Al_4_C_5_ results in the formation of Al_2_O_3_ during the oxidation process, subsequently resulting in a lowering of the viscosity of the glassy silica scale, which in turn intensifies the oxidation of the composite ceramics. This would also increase oxygen permeation through the scale, adversely affecting the oxidation resistance of the composites with high Zr_2_Al_4_C_5_ content.

## 1. Introduction

Transition-metal borides with strong covalent bond characteristics exhibit extremely high stability, hardness and strength. Among them, ZrB_2_ has excellent characteristics such as high melting point, hardness, chemical stability, thermal and electrical conductivity as well as low density, which provide it with good application prospects in the aerospace and nuclear energy industries, and in rapid cutting tools and refractory fields [1,2,3,4]. In the past two decades, ZrB_2_ has been a research hotspot in the field of ultra-high-temperature ceramics. However, the low fracture toughness of only 2~3 MPa·m^1/2^ [3,4,5] for ZrB_2_ ceramics makes it difficult to work with high strength, representing a significant challenge for its practical application.

Researchers have carried out many works on the improvement of the fracture toughness of ZrB_2_ ceramics. The addition of SiC particles into ZrB_2_ ceramics not only significantly reduced the sintering temperature, but also greatly improved the fracture toughness and high-temperature oxidation resistance in [5,6]. However, the fracture toughness of ZrB_2_-SiC composite ceramics was still low, at 4~6 MPa·m^1/2^ [5]. Compared with particle toughening, the incorporation of a larger-aspect-ratio phase into the matrix normally improves the defect size tolerance of composite materials and consumes more crack growth energy during the cracking process of composite materials, making these materials show better fracture toughness. An effective strategy is to replace SiC particles with SiC fibers or whiskers [7], as well as the further addition of toughening phases such as graphite flakes [8], graphene [9], BN [10], carbon nanotubes [11], ZrO_2_ fiber [12], and carbon fiber [13]. These toughening phases with relatively large lengths and diameters can greatly inhibit the volume effect of matrix defects and improve the toughness of the materials. Hence, materials with a large length-to-diameter ratio and excellent high-temperature mechanical properties would be optimal for toughening phases in ZrB_2_-SiC composite ceramics. A layered Zr-Al(Si)-C material obtained by simultaneously introducing Al and Si into ZrC [14], such as Zr_2_[Al_3.56_Si_0.44_]C_5_ (abbreviated Zr_2_Al_4_C_5_), showed merits such as good oxidation resistance, corrosion resistance and strong damage resistance. Together with its layered grains, high aspect ratio, good high-temperature performance and low theoretical density [15], Zr_2_Al_4_C_5_ would be an ideal toughening phase for ZrB_2_-SiC composite ceramics.

In our previous works, we successfully fabricated ZrB_2_-SiC-Zr_2_Al_4_C_5_ composite ceramics by spark plasma sintering (SPS) using the mixture of synthesized Zr_2_Al_4_C_5_ compound powders, and ZrB_2_ and SiC powders [16]. But the Zr_2_Al_4_C_5_ compound was not evenly dispersed in the ceramic matrix while the obvious agglomeration occurred, which was not conducive to the improvement of mechanical properties of ZrB_2_-SiC composite ceramics. In addition, ZrB_2_-SiC-Zr_2_Al_4_C_5_ composite ceramics were directly fabricated by reactive sintering of Zr, Al, C, B_4_C and Si powders [17], but the uneven dispersion of Zr_2_Al_4_C_5_ and the small aspect ratio of generated Zr_2_Al_4_C_5_ grains could not significantly improve the fracture toughness of composite ceramics.

In situ reaction sintering technology is an important method for sintering composite ceramics, and can ensure the dispersion uniformity of each forming phase [18,19,20]. Therefore, in this paper the Zr_2_Al_4_C_5_ compounds formed by direct in situ reaction during the sintering process were introduced into ZrB_2_-SiC composite ceramics to achieve ZrB_2_-SiC-Zr_2_Al_4_C_5_ composite ceramics with an even dispersion of Zr_2_Al_4_C_5_ grains and highly dense microstructure.

This study proves that ZrB_2_-SiC exhibits good oxidation resistance in a relatively wide temperature range [21]. However, when the temperature exceeds 1600 °C or in a low oxygen partial pressure environment, SiC will undergo active oxidation to form gaseous SiO, and the vapor pressure of molten SiO_2_ on the surface will also increase significantly, resulting in a rapid decrease in the oxidation resistance of ZrB_2_-SiC ceramics [22]. Therefore, measures such as adjusting the viscosity of the liquid protective layer, optimizing the ZrO_2_ skeleton and forming a solid protective layer have been proposed, and efforts have been made to introduce a third component addition phase to improve the oxidation resistance. The improvement of the viscosity of the liquid protective layer can further improve the high-temperature oxidation resistance of composite ceramics. To date, the substances that have been identified to effectively increase the viscosity of the SiO_2_ glass phase are mainly Ta-containing compounds, such as Ta_5_Si_3_ [23], TaSi_2_ [24] and TaB_2_ [25], etc. These additives have the same oxidation product, Ta_2_O_5_, which makes the SiO_2_ glass phase produce phase separation behavior, thereby increasing the viscosity of the glass phase and increasing the oxidation resistance of ceramics. However, when the temperature exceeds 1500 °C, the addition of Ta has a negative effect on the oxidation resistance of the material to a large extent [26], which is strongly related to the volume expansion of the Ta additive before and after the high-temperature oxidation reaction. Similarly, the addition of AlN compounds to the ZrB_2_-SiC system reduces the viscosity of the SiO_2_ glass phase, making it easier for oxygen to diffuse into the protective layer [27], making little contribution to its oxidation resistance, or even deteriorating the performance. Previous research results show that the incorporation of Zr_2_Al_4_C_5_ compounds can effectively improve the toughness of ZrB_2_-SiC composite ceramics, but there are few studies on the high-temperature oxidation properties of ZrB_2_-SiC-Zr_2_Al_4_C_5_ composite ceramics reported in the literature.

This study systematically investigates, for the first time, the microstructure, mechanical properties and oxidation behavior of ZrB_2_-SiC-Zr_2_Al_4_C_5_ composite ceramics with in situ synthesized Zr_2_Al_4_C_5_, then further explores the toughening mechanism and oxidation resistance mechanism as well as the relationship between material structure and properties, providing a theoretical basis and technical support for the research of ZrB_2_-SiC-based ultra-high-temperature ceramics.

## 2. Experimental Details

### 2.1. Preparation

The raw materials included: ZrB_2_ powder (99.5%, 4.7 μm, Alfa Aesar Co. Ltd., Ward Hill, MA, USA), SiC powder (99.5%, 5.5 μm, Shandong Weifang Huamei Company, Weifang, China), Zr powder (99.9%, 10 μm, Beijing Mengtai Youyan Technology Development Center, Beijing, China), Al powder (purity > 99.99%, 8 μm, Shanghai Chemical Reagents of Chinese Medicine Group, Shanghai, China), C powder (purity > 99.9%, 1 μm, Shanghai Capable Graphite Co. Ltd., Shanghai, China) and Si powder (purity > 99.999%, 16 μm, High Purity Materials Kojundo Chemical Laboratory Co. Ltd., Sakado, Japan).

The designed component of composite ceramics was constant 20 vol.% SiC and a total 80 vol.% of ZrB_2_ plus Zr_2_Al_4_C_5_. The Zr_2_Al_4_C_5_ grains were synthesized by the in situ reaction of Zr, Al and graphite powders (molar ratio is 2:6.2:4.8), and the Si powder (4 wt.%) was added as a sintering additive to stabilize the structure of the Zr_2_Al_4_C_5_ compounds, and is able to replace Al solid solution in Zr_2_Al_4_C_5_ to stabilize the crystal lattice [28].

ZrB_2_, SiC, Zr, Al, C and Si were weighed according to the component design, and they were mixed evenly by planetary ball mill (Vario-Planetary Mill, Fritsch Pulverisette 4, Germany) under an argon atmosphere at a speed of 400 r/min, using WC grinding media, with a ball-to-powder weight ratio of 6:1. The mixed powder slurry was dried by rotary evaporator, and then passed through a 60 mesh sieve.

The as-synthesized powders were then loaded into a graphite die with a diameter of 15 mm, and then sintered using an SPS system (model-1050, Sumitomo Coal Mining Co. Ltd., Tokyo, Japan). The temperature was measured by an optical pyrometer focused on the surface of the graphite die. The samples were heated to 600 °C at a rate of 300 °C/min, then further heated at an average heating rate of 100 °C/min to 1800 °C, at which point the temperature was held constant for 3 min. The sample was cooled naturally after the sintering period was over. A uniaxial pressure of 20 MPa and a vacuum atmosphere were applied from the start to the end of the sintering cycle.

The content of Zr_2_Al_4_C_5_ in the complex ceramics was designed as 0 vol.%, 10 vol.%, 20 vol.%, 30 vol.% and 40 vol.%, respectively, hereinafter referred to as ZAC0, ZAC10, ZAC20, ZAC30 and ZAC40.

### 2.2. Characterization and Measurement

The bulk density and open porosity of the ceramic samples were measured through Archimedes’s immersion method using water as the immersion medium. The phase compositions were analyzed by X-ray diffraction (XRD) using a Rigaku Ultima III diffractometer. Cu radiation was used and the equipment was operated at 40 KV and 40 mA. The XRD patterns were analyzed using MDI Jade v5.0 software. The microstructures of the polished surfaces and fractures were analyzed by SEM-EDS. Mean grain size was determined through image analysis on SEM micrographs of polished surfaces using a commercial software program (Image-Pro Plus v6.0, Media Cybernetics, Silver Springs, MD, USA).

The Young’s moduli (E) of the composites were determined using ultrasonic equipment (Panametrics 5072PR) with a fundamental frequency of 20 MHz, and the hardness and fracture toughness were measured using the Vickers indentation method. The indentation fracture method is an acceptable method for the estimation of the indentation fracture resistance of small ceramic products and components due to its simplicity and applicability to small test samples. The IF method does not represent the true fracture toughness, but it can be useful for describing the trend of changing fracture toughness [29,30,31]. For the detailed operation and calculation methods please refer to our previous work [32].

The oxidation resistance of the material was characterized by the following two methods:

1. The specimen was cut into a cuboid test block of 0.5 mm × 1.5 mm × 1.5 mm. After washing and drying with an ultrasonic cleaner, the TG curve and DSC curve of the test block were determined using a differential thermal analyzer (TGA, Netzsch, STA449C, Germany) during the heating process; the heating rate during the test process was 10 °C/min, and the atmosphere was air.

2. A cuboid test block with a size of 4 cm × 4 cm × 3 cm was prepared by the same method as in (1), and its surface area was expressed by *S* (unit: cm^2^). The initial mass M_0_ (unit: mg) of the test block was obtained using a high-precision balance (model: BS210S, Germany). Before oxidation, specimens were cleaned in an ultrasonic bath with alcohol. After they were dried, specimens were placed on a zirconia support with ridges to minimize the contact area and oxidation tests were conducted in an electric furnace (model Nabertherm LHT04, Germany). Specimens were heated at 5 °C/min to different temperatures and held for 30 min in static air. The mass *M*_1_ (unit: mg) of the test block was weighed again after calcination. By Formula (1), the oxidative weight gain *ΔM* per unit area of the test block was calculated.
(1)$$\Delta M=\frac{M_{1}-M_{0}}{S}$$

The phase composition and microstructure of the test block after calcination were observed and analyzed by X-ray diffractometer and scanning electron microscope.

## 3. Results and Discussion

### 3.1. Phase Composition and Microstructure

The effects of Zr_2_Al_4_C_5_ content on the density and open porosity of complex ceramics are listed in Table 1. It can be seen from the table that with the increase of the content of in situ synthesized Zr_2_Al_4_C_5_, the open porosity of the composite ceramics showed a trend of first decreasing and then increasing; when the content was 30 vol.%, the open porosity was the lowest, at only 0.22%. The density of composite ceramics showed a decreasing trend because Zr_2_Al_4_C_5_ (4.5 g/cm^2^) [15] has a lower density than ZrB_2_ (6.09 g/cm^2^) [5].

Figure 1 shows the curve of the displacement of the equipment punch with temperature during the sintering process. As can be seen from the figure, all samples shrank with increasing temperature and tended to be roughly the same, but the temperature at which ZAC30 began to shrink was relatively low, due to the addition of Zr, Al and C powders to react with oxides on the surface of the raw material at lower temperatures to eliminate impurities. At the same time, a partial glass phase was formed, which helped the sintering densification process at high temperatures [33]. In addition, the sample incorporated with Zr_2_Al_4_C_5_ began to shrink significantly at about 1400 °C because the liquid phase Al appeared during the in situ synthesized Zr_2_Al_4_C_5_ to help atom diffusion and particle rearrangement, and accelerated the sintering of composite ceramics. However, when the incorporation amount exceeded 30 vol.%, the oxide on the surface of the raw material gradually increased and the loss of Al during the sintering process was large, which affected the sintering performance of the multi-phase ceramic.

Figure 2 shows the XRD pattern of ZrB_2_-SiC-Zr_2_Al_4_C_5_ composite ceramics after sintering at 1800 °C for 3 min at different Zr_2_Al_4_C_5_ contents. As can be seen from the figure, diffraction peaks of ZrB_2_ and SiC are present for all specimens. With the increase of the content of in situ synthesized Zr_2_Al_4_C_5_, the relative intensity of the diffraction peak of Zr_2_Al_4_C_5_ gradually increases. However, when the dosage is 10 vol.%, Zr_3_Al_4_C_6_ compounds appeared because the content of Zr_2_Al_4_C_5_ was small during the sintering process, and it was easily induced by SiC to decompose at high temperatures to form Zr_3_Al_4_C_6_ compounds.

Figure 3 shows the polished surface of the samples with different Zr_2_Al_4_C_5_ contents’ backscattering electron phases. As can be seen from the figure, the black phase is SiC, the gray-white phase is ZrB_2_, and the dark gray phase is the Zr_2_Al_4_C_5_ layered compound. Compared with the direct incorporation of Zr_2_Al_4_C_5_ layered compounds [15,16], the Zr_2_Al_4_C_5_ compounds generated by in situ reaction could be dispersed in the ZrB_2_-SiC ceramic matrix relatively uniformly, and there was no obvious agglomeration. With the increase of Zr_2_Al_4_C_5_ content, the in situ synthesized Zr_2_Al_4_C_5_ layered compounds gradually increased, the length–diameter ratio also gradually increased and the ZrB_2_ grain size gradually decreased. When the content was 30 vol.%, the ZrB_2_ grain size was about 3.3 μm (as shown in Table 1), which indicated that the in situ reaction to generate Zr_2_Al_4_C_5_ compounds could inhibit the growth of ZrB_2_ grains; however, when the content was 40 vol.%, abnormal growth occurred, which was not conducive to the improvement of mechanical properties.

### 3.2. Mechanical Properties

The effects of different Zr_2_Al_4_C_5_ contents on the mechanical properties of multi-phase ceramics are summarized in Table 1. It can be seen from the table that with the increase of Zr_2_Al_4_C_5_ content, the Young’s modulus and Vickers hardness of the multi-phase ceramics gradually decreased because the Young’s modulus and Vickers hardness of Zr_2_Al_4_C_5_ are smaller than those of ZrB_2_ [5,15]. With the increase of Zr_2_Al_4_C_5_ content, the fracture toughness first increased and then decreased, and when the dosage was 30 vol.%, its fracture toughness reached a maximum of 5.26 MPa·m^1/2^, which represents an increase of about 30% compared with ZrB_2_-SiC ceramics, and an increase of nearly 20% compared with that obtained by the direct incorporation of Zr_2_Al_4_C_5_ [16], because the in situ synthesized Zr_2_Al_4_C_5_ compound can be evenly dispersed in the ZrB_2_ matrix. Additionally, the length–diameter ratio also increased and the probability of encountering layered structures during crack propagation was also therefore increased, causing more grains to be pulled out or fractured along the crystal and leading to an increase in the crack propagation energy that needs to be consumed. Thereby, the fracture toughness of the material was increased.

Figure 4 shows the SEM fracture surface morphologies of samples at different Zr_2_Al_4_C_5_ contents. It can be seen the ZrB_2_ grain size of the ZAC0 sample was large, as shown in Figure 4a, and the fracture of the complex ceramic was relatively flat. When the Zr_2_Al_4_C_5_ compound was generated by in situ reaction, the fracture of the complex ceramic was rough and dense, and there were traces left by the dialing out of the layered Zr_2_Al_4_C_5_ compound. The fracture mode was presented as a mixed fracture mode of inter- and trans-granular fractures. With the increase of Zr_2_Al_4_C_5_ content, the roughness degree increased and the grain size of ZrB_2_ gradually decreased, as shown in Table 1.

Figure 5 shows the crack tip propagation path diagrams of samples when measuring Vickers hardness. It can be seen that when the Zr_2_Al_4_C_5_ compound was incorporated to change the crack propagation path, there was obvious crack shift and crack bridging (Figure 5b,c). At the same time, when the crack encountered Zr_2_Al_4_C_5_ during the propagation process, the crack repeatedly shifted between its layers and expanded along the weak interface between layers, as shown in Figure 5c, which consumed crack growth energy and improved the fracture toughness of the multi-phase ceramics [34].

### 3.3. Oxidation Resistance

Figure 6 shows the change of oxidative weight gain and oxide layer thickness of samples under different Zr_2_Al_4_C_5_ contents. It can be seen from the Figure 6 that with the increase of oxidation temperature, the oxidative weight gain and oxide layer thickness of all samples showed a gradually increasing trend; when the temperature was higher than 1200 °C, the oxidative weight gain and oxide layer thickness increase trend was more obvious. With the increase of Zr_2_Al_4_C_5_ content, the oxidative weight gain and oxide layer thickness showed a trend of first increasing, then decreasing and then increasing, and when the dosage was 30 vol.%, the oxidative weight gain was the smallest—16.2 mg/cm^2^ at 1500 °C. However, previous research has found that when the content reaches 40 vol.%, the oxidation of complex ceramics is intensified because the Zr_2_Al_4_C_5_ compound is oxidized to generate Al_2_O_3_ and SiO_2_ to form aluminosilicates, which in turn leads to a decrease in the glass phase content of SiO_2_, and the glass phase cannot uniformly cover the surface of the oxide layer, accelerating the diffusion of oxygen to the interior [35]. In addition, more gaseous volatile substances are formed after oxidation, and the evolution of gas causes the oxide layer to form a porous structure, and the Al_2_O_3_ and ZrO_2_ generated by oxidation are dissolved in the glass phase to form a thick and loose oxide layer, which provides many paths for oxygen atoms to enter the inside of the material, causing more serious oxidation.

Figure 7 shows the thermal stability of ZAC30 and ZAC40 by thermogravimetric (TG) analysis and differential scanning calorimetry (DSC). From the DSC curve, it can be seen that the initial oxidation temperatures of ZAC30 and ZAC40 were 772.5 °C and 607.5 °C, respectively, which indicates that the addition of 30 vol.% Zr_2_Al_4_C_5_ can delay the occurrence of oxidation in the early stage of oxidation. In contrast to the observations of ZAC30, ZAC40 had a significant exothermic peak at 1446.5 °C, which was caused by the reaction of SiO_2_ and Al_2_O_3_ and ZrO_2_ by oxidation to form aluminosilicate and ZrSiO_4_. It can be seen from the TG curve that with the increase of oxidation temperature, oxidative weight gain gradually increased, and at temperatures above 1200 °C, the slope of the TG curve increased significantly and the weight gain was obvious.

Figure 8 shows the XRD patterns of the surface layer of the composite ceramic after oxidation for 30 min at different temperatures. The diffraction peaks of ZrB_2_, SiC and Zr_2_Al_4_C_5_ compounds could be detected at an oxidation temperature of 800 °C, but there were ZrO_2_ diffraction peaks at the same time because ZrB_2_ and a small amount of Zr_2_Al_4_C_5_ compounds were oxidized to form ZrO_2_ and Al_2_O_3_ compounds. When the oxidation temperature rose to 1000 °C, the diffraction peak of ZrO_2_ gradually strengthened, while the diffraction peak of Zr_2_Al_4_C_5_ was not detected, but the diffraction peak of Al_2_O_3_ was observed. As the temperature continued to rise, the diffraction peaks of ZrO_2_ and Al_2_O_3_ gradually increased, and when the oxidation temperature reached 1400 °C, the aluminosilicate diffraction peak was significantly enhanced, and with the increase of Zr_2_Al_4_C_5_ content, the diffraction peak gradually weakened. At 1500 °C, the main phases on the surface of the oxide layer were ZrSiO_4_, ZrO_2_ and aluminosilicate, which were due to the reaction of SiO_2_ with ZrO_2_ to form ZrSiO_4_ compounds, and SiO_2_ with Al_2_O_3_ to form aluminosilicates. When the volume content was 30 vol.%, the peak strength of ZrSiO_4_ and aluminosilicate diffraction peaks was the strongest.

Figure 9 shows SEM pictures of the cross section and surface of samples after oxidation at 1200 °C for 30 min. It can be seen the oxide layer of the ZAC30 sample was the thinnest, with a thickness of 40 μm, and a thin glassy substance appeared on the surface, which hindered the diffusion of oxygen atoms. However, there were obvious holes in the oxide layer of the ZAC40 sample. It can be seen from the SEM diagram on the surface that the surface of the complex ceramic was uneven and glassy after oxidation, but there were obvious cracks on the surface of ZAC40.

Figure 10 shows SEM and EDS pictures of the cross section and surface of the samples after oxidation at 1500 °C for 30 min. It can be seen from the cross-sectional view that the thickness of the oxide layer showed a trend that first increased then decreased with increasing Zr_2_Al_4_C_5_ content; the composite ceramic with 30 vol.% Zr_2_Al_4_C_5_ showed the thinnest oxide layer. When the volume content was 10 vol.%, the oxide layer was the thinnest, only 170 μm, and the thicker glass phase formed on the surface effectively prevented the diffusion of oxygen atoms. However, when the content increased to 20 vol.%, the glass phase of the surface layer gradually disappeared because the formation of Al_2_O_3_ after oxidation reduced the viscosity of the glass phase and the low-viscosity glass phase also allowed easy penetration of the oxide layer, resulting in no obvious stratification of the oxide layer while the low-viscosity glass phase accelerated the mass transfer process, resulting in the growth of ZrO_2_ grains. When the volume content was increased to 30 vol.%, the oxide layer section had obvious stratification and the oxide layer was thinner, 235 μm, and there was also a thick glass phase on the surface, which effectively prevented oxygen atoms from diffusing into the matrix. When the incorporation amount was 40 vol.%, there was no obvious delamination, the thickness of the glass phase on the surface of the oxide layer progressively decreased, and a large number of holes appeared in the oxide layer, showing a loose porous structure, because with the increase of Zr_2_Al_4_C_5_ content, the content of Al_2_O_3_ generated after oxidation gradually increased, and Al_2_O_3_ and SiO_2_ generated aluminosilicate, resulting in a decrease in the glass phase content of SiO_2_, which could not be uniformly covered on the surface of the oxide layer. Therefore, only by controlling the appropriate ratio of Al to Si can the oxidation resistance of composite ceramics be optimal because the ratio of Al to Si controls the glass phase viscosity, which can form a dense glass phase protective layer.

It can be seen from its surface diagram that when the volume content was 10 vol.%, the surface of the oxide layer presented a long strip-like crystalline phase; from the XRD and EDS analysis it can be seen that these crystals were aluminosilicate and Zr_2_SiO_4_. With the increase of Zr_2_Al_4_C_5_ content, the surface of the oxide layer showed obvious unevenness, and the integrity of the surface oxide film was destroyed, indicating that a violent oxidation reaction occurred at this time, generating a large number of volatile gaseous products, which escaped the surface and left holes. Combined with EDS analysis, it can be seen that the main elements on the surface of the oxide layer were silicon, aluminum, oxygen, aluminosilicate and SiO_2_ glass phase. With the increase of Zr_2_Al_4_C_5_ volume content, the O content gradually increased.

The distribution of Zr, Al and Si elements can be clearly seen in the cross-sectional element surface scan of the ceramic after oxidation, as shown in Figure 11. It can be seen that when the oxidation temperature was 1200 °C, Al elements gathered on the surface of the oxide layer while silicon elements were present in the lower layer of Al elements. As the diffusivity of Al is greater than the diffusivity of Si, Al more easily diffuses outward at high temperatures [36], and Al_2_O_3_ formed by oxidation at this temperature is more present in the surface layer of oxidation. When the oxidation temperature was 1400 °C, Si elements began to migrate to the surface of the oxide layer, obvious missing layers of Si and Al elements appeared, and Al_2_O_3_, SiO_2_ and aluminosilicate generated after oxidation of the complex ceramics covered the surface of the oxide layer. When the oxidation temperature rose to 1500 °C, the Zr element basically did not appear on the surface of the oxide layer; at this time, due to the high temperature, the Al element on the surface of the oxide layer was also reduced, and the Si element mainly existed on the surface of the oxide layer, so that the surface of the oxide layer was mainly the glass phase of SiO_2_ and Al-Si-O, thereby preventing the diffusion of oxygen atoms. Combined with XRD and EDS analysis, the section area after oxidation was divided into four layers: the first layer was the surface of the oxide layer, namely SiO_2_, aluminosilicate and Zr_2_SiO_4_ and a small amount of ZrO_2_; the second layer (i.e., the lower layer of the oxide layer) was ZrO_2_, Al_2_O_3_, SiO_2_ and C; the third layer was the missing layer of SiC and Zr_2_Al_4_C_5_ compounds, at which time SiC and Zr_2_Al_4_C_5_ were oxidized to the surface layer at high temperatures (e.g., ZrO_2_, SiO_2_ and Al_2_O_3_); the fourth layer was the unreacted complex ceramic region.

Figure 12 shows a BESEM picture of the ZAC30 sample after 30 min oxidation at 1500 °C. Figure 12b is a local magnification of the second layer area shown in Figure 12a; through EDS analysis, it can be seen that the peaks of Zr, Al and Si appeared at point 1, and the glass phase of Al-Si-O was generated there; the peaks of Si and O at point 3 were more obvious, and the SiO_2_ glass phase was generated here; the bright phase was determined to be ZrO_2_ by EDS. Through EDS analysis, it was found that the composition of the glass phase at point 1 and point 3 was different, which was because the diffusivity of Al is greater than that of Si, and Al is more likely to diffuse to the surface layer at high temperature—that is, the surface layer formed a glass phase with a high content of Al-Si-O. Figure 12c shows the area where the third and fourth layers met, and from the enlarged view, it can be seen that the layered Zr-Al-C compounds were the first oxidized, and the layered structure appeared petal-like after oxidation, following the formation of ZrO_2_ and Al_2_O_3_ compounds.

## 4. Conclusions

ZrB_2_-SiC-Zr_2_Al_4_C_5_ multi-phase ceramics with uniform structure and high density were successfully produced by an in situ reaction sintering process. With the increase of Zr_2_Al_4_C_5_ volume content, the open porosity of the composite ceramics showed a trend of first decreasing and then increasing, and when the volume content of the in situ synthesized Zr_2_Al_4_C_5_ was 30 vol.%, the porosity of the complex ceramic reached its minimum. Moreover, the grain size of ZrB_2_ showed a trend of gradual reduction.

Both the Vickers hardness and Young’s modulus of composite ceramics gradually decreased with increasing Zr_2_Al_4_C_5_ volume content. The fracture toughness showed a trend of first increasing and then decreasing, and the fracture toughness increased by about 30% compared with that of ZrB_2_/SiC ceramics. It presented as a combination fraction mode of inter- and trans-granular fracture, and there was a phenomenon of pulling out and bridging of layered Zr_2_Al_4_C_5_ compounds, which was conducive to the improvement of their fracture toughness.

The oxidative weight gain showed a trend of first increasing and then decreasing and then increasing with an increase in the Zr_2_Al_4_C_5_ volume content. When the volume content was 30 vol.%, the oxidative weight gain was the smallest, and there was a thicker glass phase on the surface after oxidation at 1500 °C. However, when the volume content was 20 vol.% and 40 vol.%, the glass phase on the surface of the oxide layer disappeared but obvious holes were present. This is because Zr_2_Al_4_C_5_ compounds were oxidized to generate Al_2_O_3_ and SiO_2_ to form aluminosilicates, which in turn led to a decrease in the content of SiO_2_ glass. The glass phase could not uniformly cover the surface of the oxide layer while more gaseous volatile substance was generated, which provided more paths for oxygen atoms to enter the material and intensified the oxidation of the composite ceramics.

Combined with XRD and EDS analysis, the sample after oxidation could be divided into four layers when the volume content was 30 vol.%: the first layer was the surface of the oxide layer, which included SiO_2_, aluminosilicate and Zr_2_SiO_4_ and a small amount of ZrO_2_; the second layer was ZrO_2_, Al_2_O_3_, SiO_2_ and C; the third layer was the missing layer of SiC and Zr_2_Al_4_C_5_ compounds; and the fourth layer was an unreacted complex ceramic region.

## Figures and Tables

**Figure 1 materials-16-04495-f001:**
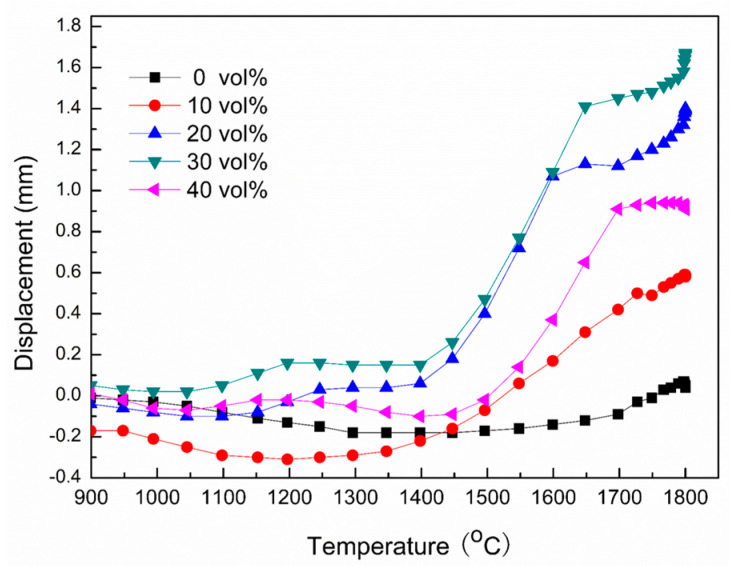
Displacement curve of pressure punch at different temperatures.

**Figure 2 materials-16-04495-f002:**
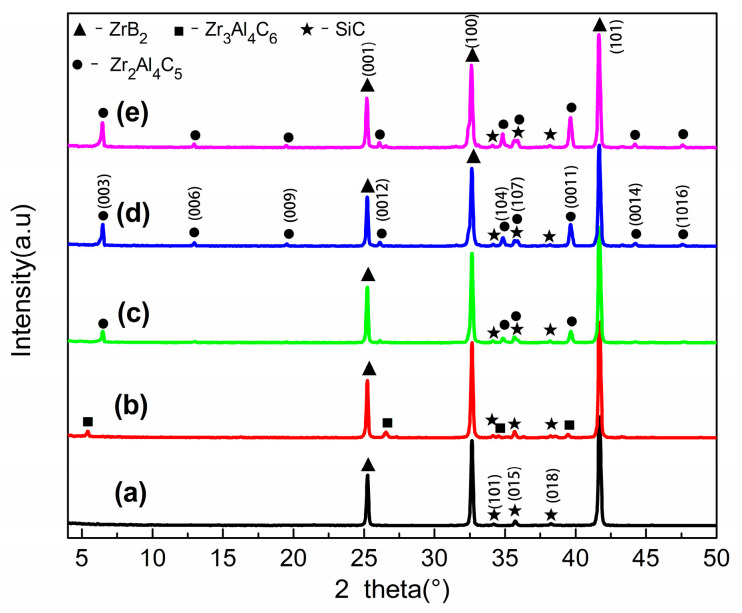
XRD patterns of composite ceramics at different Zr_2_Al_4_C_5_ contents: (**a**) ZAC0, (**b**) ZAC10, (**c**) ZAC20, (**d**) ZAC30, (**e**) ZAC40.

**Figure 3 materials-16-04495-f003:**
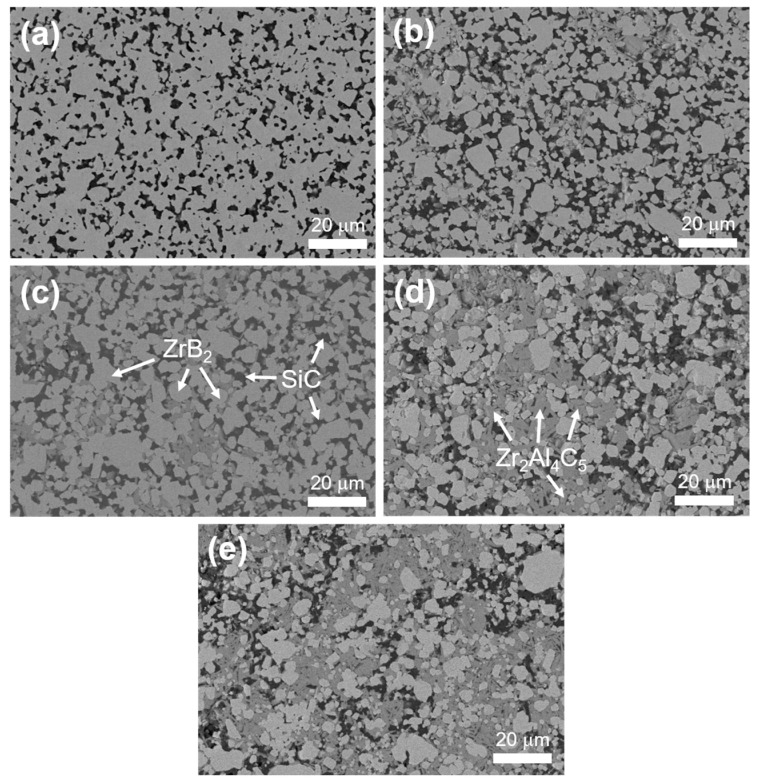
Backscattered electron phase pictures of polished surfaces of multi-phase ceramics at different Zr_2_Al_4_C_5_ contents: (**a**) ZAC0, (**b**) ZAC10, (**c**) ZAC20, (**d**) ZAC30, (**e**) ZAC40.

**Figure 4 materials-16-04495-f004:**
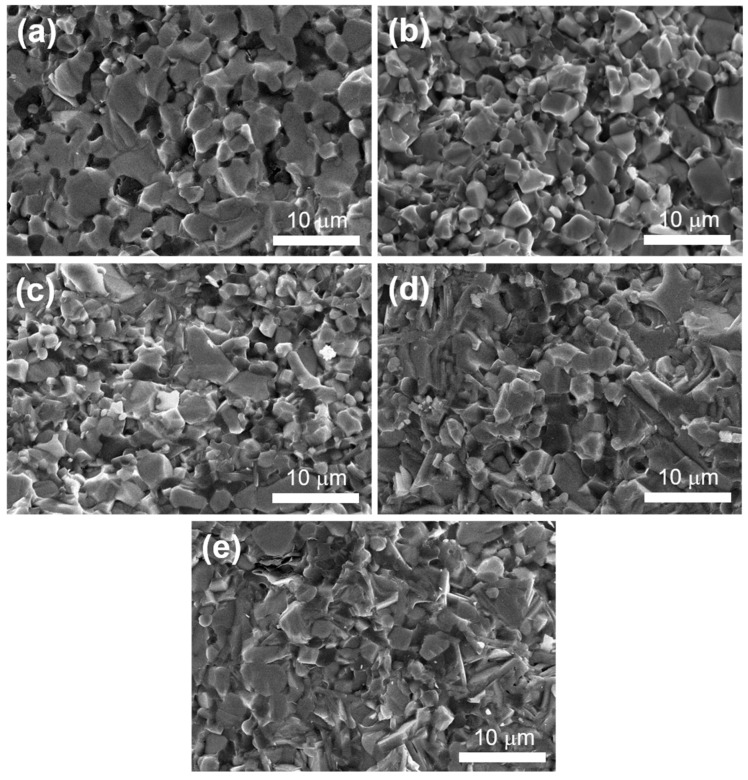
SEM fracture surface morphologies of samples with different Zr_2_Al_4_C_5_ content: (**a**) ZAC0, (**b**) ZAC10, (**c**) ZAC20, (**d**) ZAC30, (**e**) ZAC40.

**Figure 5 materials-16-04495-f005:**
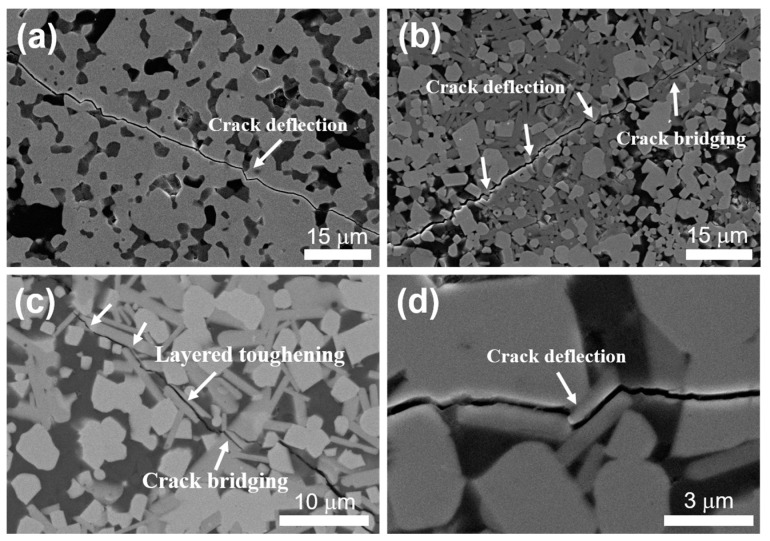
Expansion path diagram of the crack tip of multi-phase ceramics: (**a**) ZAC0, (**b**) ZAC20, (**c**) ZAC30, (**d**) ZAC30.

**Figure 6 materials-16-04495-f006:**
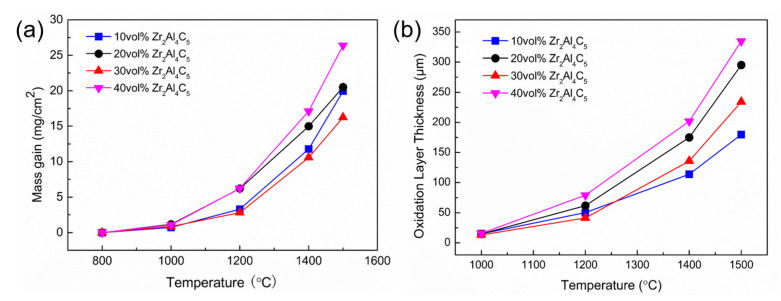
(**a**) Changes of oxidative weight gain and (**b**) oxidation layer thickness of samples at different contents of Zr_2_Al_4_C_5_.

**Figure 7 materials-16-04495-f007:**
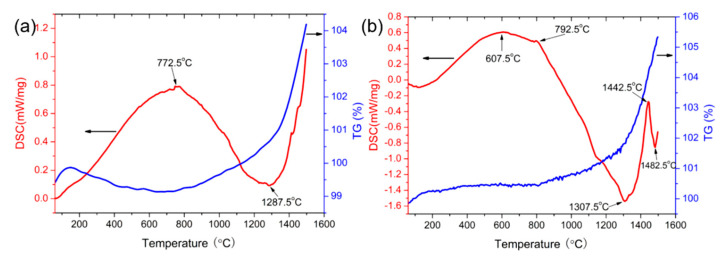
TG–DSC curve of multi-phase ceramics: (**a**) ZAC30, (**b**) ZAC40. Red line is differential scanning calorimetry, blue line is thermogravimetry.

**Figure 8 materials-16-04495-f008:**
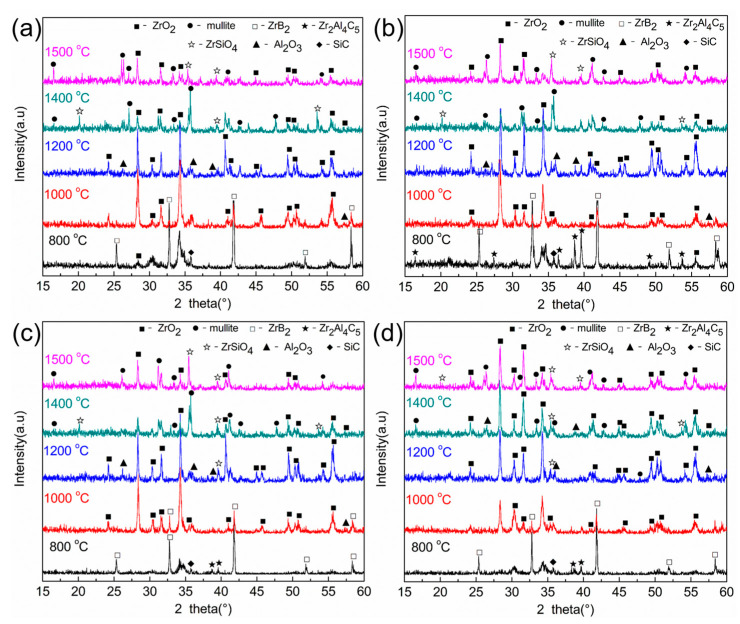
XRD patterns of composite ceramics oxidized for 30min at different oxidation temperatures: (**a**) ZAC10, (**b**) ZAC20, (**c**) ZAC30, (**d**) ZAC40.

**Figure 9 materials-16-04495-f009:**
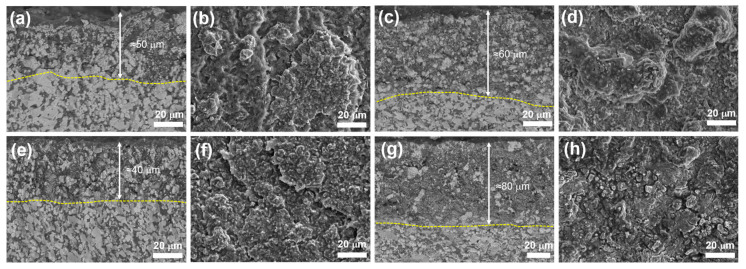
SEM pictures of the cross section and surface of the samples after oxidation at 1200 °C for 30 min: (**a**,**b**) ZAC10, (**c**,**d**) ZAC20, (**e**,**f**) ZAC30, (**g**,**h**) ZAC40.

**Figure 10 materials-16-04495-f010:**
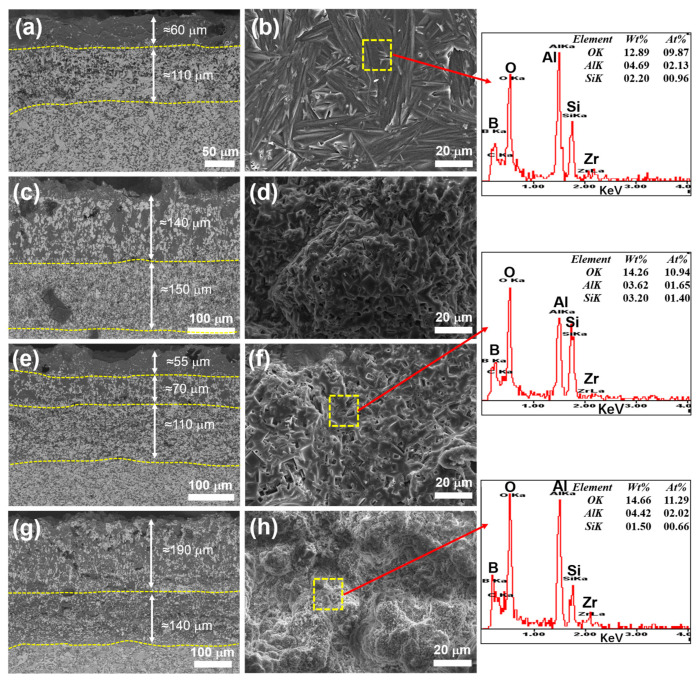
SEM and EDS pictures of the cross section and surface of the samples after oxidation at 1200 °C for 30 min: (**a**,**b**) ZAC10, (**c**,**d**) ZAC20, (**e**,**f**) ZAC30, (**g**,**h**) ZAC40.

**Figure 11 materials-16-04495-f011:**
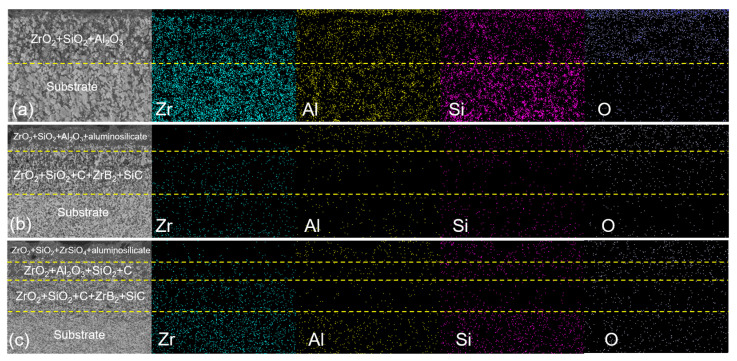
EDS pattern of the ZAC30 sample at different oxidation temperatures: (**a**) 1200 °C, (**b**) 1400 °C, (**c**) 1500 °C.

**Figure 12 materials-16-04495-f012:**
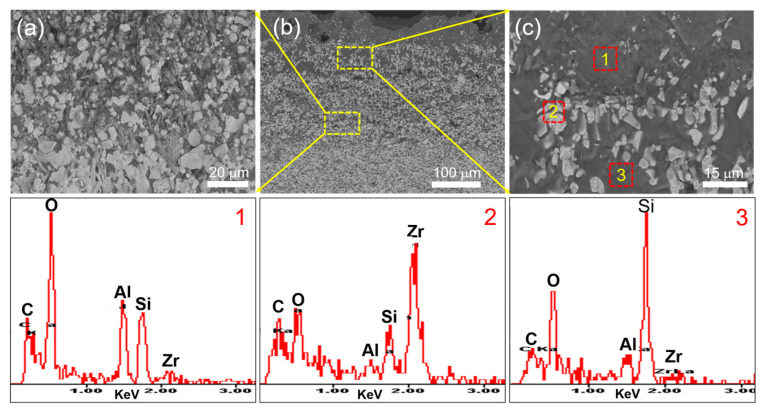
BESEM and EDS micrographs of the oxide layer section of the ZAC30 specimen after oxidation at 1500 °C for 30 min (**a**–**c**). (**a**,**c**) magnified SEM images of the areas enclosed in yellow dotted rectangles in (**b**).

**Table 1 materials-16-04495-t001:** Effects of Zr_2_Al_4_C_5_ content on density, open porosity, ZrB_2_ grain size and mechanical properties of composite ceramics.

Sample	Density (g/cm^3^)	Open Porosity (%)	ZrB_2_ Grain Size (μm)	Vickers Hardness (GPa)	Young’s Modulus (GPa)	Fracture Toughness (MPa·m^1/2^)
ZAC0	5.36	0.65	4.7 ± 0.7	17.9 ± 0.2	462 ± 15	4.02 ± 0.30
ZAC10	5.07	0.45	4.2 ± 0.6	17.1 ± 0.2	435 ± 20	4.61 ± 0.33
ZAC20	4.96	0.29	3.9 ± 0.6	16.8 ± 0.2	407 ± 15	5.12 ± 0.29
ZAC30	4.83	0.22	3.3 ± 1.0	16.6 ± 0.4	392 ± 15	5.26 ± 0.32
ZAC40	4.63	0.41	3.7 ± 1.1	15.8 ± 0.4	376 ± 15	4.53 ± 0.30

## Data Availability

Not applicable.

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
