# Peer review of "Effect of Zr_2_Al_4_C_5_ Content on the Mechanical Properties and Oxidation Behavior of ZrB_2_-SiC-Zr_2_Al_4_C_5_ Ceramics"

_materials, 2023, doi:10.3390/ma16124495_

Round 1

Reviewer 1 Report

Manuscript ID: materials-2429472

Dear Editor,

The authors used SPS technique to in-situ produce Zr2Al4C5 in the ZrB2-SiC matrix. The mechanical properties (Vickers hardness, fracture toughness, young modulus), microstructure, oxidation behaviour and physical property (density) were investigated.

The overall quality of the submitted manuscript is excellent, only using IF technique and correcting the term is considered a critical issue as IF technique for measuring fracture toughness of ceramic materials is questionable. It can be used just describe the trend not taking the actual values.

1-The Abstract section is well presented.

2-Introduction section: is well arranged and completely understandable to readers.

3-The critical point starts with the preparation section: are the powders mixed in vol. % or wt. %? the authors should be consistent in this matter.

4-The equations should be rewritten with mathtype.

5-for fracture toughness, Indentation fracture (IF) method is questionable.

Even if consider for this fact that using this technique by the authors is acceptable due to the difficulty using more advanced techniques such as SEVNB,...etc, the authors need to use the term of Indentation Fracture resistance (KIFR) Instead of fracture toughness KIC. This is not a real fracture toughness, this IF technique produces a fracture toughness that only its trend of increasing and decreasing can be useful for describing the trend of changing this mechanical properties upon an external stimuli, not taking its actual values.

This fact, for example, was explained in detail in the following articles. And the authors are required to correct this term based on explanation in these references.

https://doi.org/10.1016/j.jeurceramsoc.2017.07.027

https://doi.org/10.1016/j.ijrmhm.2018.03.006

6-Figure 2 needs to have a higher resolution.

7-Line 185: remove a picture... this is not used in academic manuscripts.

8-mechanical properties accompanied with SEM micrographs is well presented and explained, especially for what called fracture toughness (actually (KIFR)).

9-XRD figure (8) needs to be provided in higher resolution.

10-usually a distance should be kept between a value and its measuring unit. 40 HV, 40 vol.%, ...etc (as an example).

11-The authors check whether the reference style is in online with the Journal of Materials ?

Author Response

Dear reviewer,

We greatly appreciate your valuable comments and great suggestions. After careful discussion, we would like to submit our revision with the detailed response to the questions as followings.

1.The critical point starts with the preparation section: are the powders mixed in vol. % or wt. %? the authors should be consistent in this matter.

Thank you for the comments and suggestions. The Si powder (4 wt%) was added as a sintering additive for stabilizing the structure of Zr2Al4C5 compounds, 4 wt% of the total weight of Zr, Al and C powders. Therefore we use the wt.%.

2. The equations should be rewritten with mathtype.

Thank you for the comments and suggestions. We revised the equations with mathtype.

3. for fracture toughness, Indentation fracture (IF) method is questionable.

Even if consider for this fact that using this technique by the authors is acceptable due to the difficulty using more advanced techniques such as SEVNB,...etc, the authors need to use the term of Indentation Fracture resistance (KIFR) Instead of fracture toughness KIC. This is not a real fracture toughness, this IF technique produces a fracture toughness that only its trend of increasing and decreasing can be useful for describing the trend of changing this mechanical properties upon an external stimuli, not taking its actual values.

This fact, for example, was explained in detail in the following articles. And the authors are required to correct this term based on explanation in these references.

https://doi.org/10.1016/j.jeurceramsoc.2017.07.027

https://doi.org/10.1016/j.ijrmhm.2018.03.006

Thank you for the comments and suggestions. We use the term of Indentation Fracture resistance, this is because size of the sample is too small. We added a description on the effect of Indentation Fracture resistance in the Experimental details: “The indentation fracture method is an acceptable method for estimation of indentation fracture resistance of small ceramic products and components due to its simplicity and applicability to small test samples. IF method does not represent the true fracture toughness, but it can be useful for describing the trend of changing the fracture toughness[27-28].”

4. Figure 2 needs to have a higher resolution.

Thank you for the comments and suggestions. We revised the figure 2.

5. Line 185: remove a picture... this is not used in academic manuscripts.

Thank you for the comments and suggestions. We revised this sentence into “Fig.3 shows the polished surface of the samples with different Zr2Al4C5 contents backscattering electron phases.”

6. XRD figure (8) needs to be provided in higher resolution.

Thank you for the comments and suggestions. We revised the figure 8.

7. usually a distance should be kept between a value and its measuring unit. 40 HV, 40 vol.%, ...etc (as an example).

Thank you for the comments and suggestions. We revised the distance.

8. The authors check whether the reference style is in online with the Journal of Materials ?

Thank you for the comments and suggestions. We revised the reference style.

We have rearranged text body and marked all revisions in red. Hopefully the revision will be satisfied.

Sincerely,

Reviewer 2 Report

The aim of the manuscript is systematically investigated and present the microstructural, mechanical properties, and oxidation behavior of the ZrB2-SiC-Zr2Al4C5 composite. The materials were synthesized using the SPS method and characterized with the application of XRD, SEM, DSC, and TG methods. All the obtained results are interpreted correctly and draw conclusions are justified by the results. The manuscript is well-written and may be published after considering the following minor remark:

What are the uncertainty of the density and open porosity (Table 1) and results in Fig.6

Author Response

Dear reviewer,

We greatly appreciate your valuable comments and great suggestions. After careful discussion, we would like to submit our revision with the detailed response to the questions as followings.

What are the uncertainty of the density and open porosity (Table 1) and results in Fig.6

Thank you for the comments and suggestions. We specifically indicated the Table 1 and Fig. 6. We revised “0.29%” into “0.22%” at Lines 157. We revised “4.15 g/cm2” into “4.5 g/cm2” of density of Zr2Al4C5 composite at Lines 158.

We have rearranged text body and marked all revisions in red. Hopefully the revision will be satisfied.

Sincerely

Reviewer 3 Report

Getting acquainted with the manuscript entitled “Effect of Zr2Al4C5 content on the mechanical properties and oxidation behavior of ZrB2-SiC-Zr2Al4C5 ceramics” I would provide the following comments to its authors.

1.     Lines 111-112:

The content of SiC in ZrB2-SiC-Zr2Al4C5 composite ceramics is fixed at 20vol.%, of 111

which Zr2Al4C5 is synthesized by in situ reaction of Zr powder, Al powder and C powder. 112

Why “of which”? The sense would be clearer when the sentence divided in two - the first about the amount of SiC and the second - about the way Zr2Al4C5 was synthesized.

The text written in lines 111-116 is difficult to comprehend. Please simplify the sentences.

2.     The graph in Figure 1 should be constructed as Displacement vs. Temperature as the authors discussed this dependence in the relevant text (lines 162-174).

3.     Figure 2 and the relevant text (lines 178-184):

-        XRD patterns visualization is very poor, some of the marked peaks are invisible (for example, in the pattern (b));

-        Not all the observed peaks were attributed to any phase (for example, between 30-35° 2theta in the pattern (e));

-        Miller indices were not shown;

-        The authors claimed that “As can be seen from the figure, diffraction peaks of ZrB2 and SiC are present in all specimens”. However, SiC peaks were marked only for the pattern (a)

-        No details on the phase analysis were presented in the Experimental section. Which data was used as a reference for each phase, which database it was taken from?

-        The authors claimed that “With the increase of the content of in-situ synthesized Zr2Al4C5, the relative intensity of the diffraction peak of Zr2Al4C5 gradually increases”. Which peak did they mean and how could we estimate its changes, if on the one hand, there were no reference phase (for instance, a constant amount of crystalline silica or alumina in each studied sample), and on the other hand the patterns were not fully shown (please see the paeks of ZrB2 in the pattern (e)). To compare phase contents of the prepared samples correctly, quantitative phase analysis based on these XRD patterns should be provided.

4.     In the lines 213-214, the authors concluded that the fracture toughness was improved by the addition of Zr2Al4C5 to ZrB2-SiC material. Only based on the data from the table 1, it was. But in the Introduction, it was noted that fracture toughness of ZrB2-SiC is about 4-6 MPa*m1/2. The obtained 5.26 MPa*m1/2, which was the highest value among the samples prepared by the authors, lies in this range. I suppose this needs closer discussion.

5.     Lines 262-264:

ZAC40 has a significant exothermic peak at 1446.5 °C, which is caused by the reaction of SiO2 and Al2O3 and ZrO2 by oxidation to form aluminosilicate and ZrSiO4.

6.     In Figure 7a, why did the TG curve start not from 100%?

7.     Figure 8: in XRD patterns, a phase cannot be correctly detected based on less than three experimental peaks matching with the reference data. Again, which database was employed for the analysis?

8.     Figure 10: EDX spectra and the related data are not readable.

9.     Lines 301-302: “shows a trend of first increasing and then decreasing and increasing” - please express this more analytically.

10.  In Figure 11, brightness and/or contrast must be corrected, because the images look almost black giving no chance to distinguish the element’s mapping.

11.  In Figure 12, EDX spectra are too small for perception.

To summarize, the authors performed quite a lot of experiments to study the properties of the manufactured multiphase ceramics, but the obtained results are difficult to be estimated because of mostly poor visualization and interpretation. There are also serious concerns on the practical importance of these results as the mechanical properties of the ceramics were slightly improved at best (fracture toughness) or even deteriorated (Young’s modulus, Vickers hardness) compared to the parent material (ZrB2-SiC). Besides, the soundness of the results is not quite clear because the authors didn’t provide their comparison to the latest literature data.

English must be revised. Quite often the construction of the sentenses made them difficult to be understood.

Author Response

Dear reviewer,

We greatly appreciate your valuable comments and great suggestions. After careful discussion, we would like to submit our revision with the detailed response to the questions as followings.

1. Lines 111-112: The content of SiC in ZrB2-SiC-Zr2Al4C5 composite ceramics is fixed at 20 vol.%, of 111 which Zr2Al4C5 is synthesized by in situ reaction of Zr powder, Al powder and C powder. 112 Why “of which”? The sense would be clearer when the sentence divided in two - the first about the amount of SiC and the second - about the way Zr2Al4C5 was synthesized. The text written in lines 111-116 is difficult to comprehend. Please simplify the sentences.

Thank you for the comment. We shortened the sentences. We revised the sentence into: “The designed component of composite ceramics is constant 20 vol% SiC and a total 80 vol% of ZrB2 plus Zr2Al4C5. The Zr2Al4C5 grains are synthesized by in situ reaction of Zr, Al and graphite powders (molar ratio is 2:6.2:4.8), and the Si powder (4 wt%) is added as a sintering additive for stabilizing the structure of Zr2Al4C5 compounds, and can replace Al solid solution into Zr2Al4C5 to stabilize the crystal lattice [28].”

2. The graph in Figure 1 should be constructed as Displacement vs. Temperature as the authors discussed this dependence in the relevant text (lines 162-174).

   Thank you for the comment and suggestion. We revised the figure 1 as Displacement vs. Temperature.

3. Figure 2 and the relevant text (lines 178-184): XRD patterns visualization is very poor, some of the marked peaks are invisible (for example, in the pattern (b)); Not all the observed peaks were attributed to any phase (for example, between 30-35° 2 theta in the pattern (e)); Miller indices were not shown;

Thank you for the comment and suggestion. We revised the figure 2.

The authors claimed that “As can be seen from the figure, diffraction peaks of ZrB2 and SiC are present in all specimens”. However, SiC peaks were marked only for the pattern (a)

Thank you for the comment and suggestion. We added diffraction peaks of ZrB2 and SiC in the Figure 2.

No details on the phase analysis were presented in the Experimental section. Which data was used as a reference for each phase, which database it was taken from?

Thank you for the comment and suggestion. We added an explanation in the Experimental details: “The XRD patterns were analyzed using MDI Jade 5.0 software.”

The authors claimed that “With the increase of the content of in-situ synthesized Zr2Al4C5, the relative intensity of the diffraction peak of Zr2Al4C5 gradually increases”. Which peak did they mean and how could we estimate its changes, if on the one hand, there were no reference phase (for instance, a constant amount of crystalline silica or alumina in each studied sample), and on the other hand the patterns were not fully shown (please see the paeks of ZrB2 in the pattern (e)). To compare phase contents of the prepared samples correctly, quantitative phase analysis based on these XRD patterns should be provided.

Thank you for the comment and suggestion. We revised the figure 2, and can see the peaks of ZrB2 in the pattern (e). We have taken this into account, but the XRD patterns of ceramic samples were recorded at a scan speed of 5 ◦/min. We can't provide quantitative phase analysis based on these XRD patterns.

4. In the lines 213-214, the authors concluded that the fracture toughness was improved by the addition of Zr2Al4C5 to ZrB2-SiC material. Only based on the data from the table 1, it was. But in the Introduction, it was noted that fracture toughness of ZrB2-SiC is about 4-6 MPa*m1/2. The obtained 5.26 MPa*m1/2, which was the highest value among the samples prepared by the authors, lies in this range. I suppose this needs closer discussion.

Thank you for the comment and suggestion. This is because the raw materials, preparation methods and test methods used by each scholar are different in the literature, which leads to the fracture toughness of the literature is only a range value. But in this paper, the fracture toughness of ZrB2-SiC-Zr2Al4C5 composite was increased by about 30 %, compared with ZrB2-SiC ceramics. This proves that in-situ synthesized Zr2Al4C5 layered compounds can improve the fracture toughness of ZrB2-SiC ceramics.

5. Lines 262-264: ZAC40 has a significant exothermic peak at 1446.5 °C, which is caused by the reaction of SiO2 and Al2O3 and ZrO2 by oxidation to form aluminosilicate and ZrSiO4.

Thank you for the comment and suggestion. We specifically indicated the sentences.

6. In Figure 7a, why did the TG curve start not from 100%?

Thank you for the comment and suggestion. We think that this may be experimental errors. We will continue to study it.

7. Figure 8: in XRD patterns, a phase cannot be correctly detected based on less than three experimental peaks matching with the reference data. Again, which database was employed for the analysis?

Thank you for the comment and suggestion. We revised the figure 8.

8. Figure 10: EDX spectra and the related data are not readable.

   Thank you for the comment and suggestion. We revised the figure 10.

9. Lines 301-302: “shows a trend of first increasing and then decreasing and increasing” - please express this more analytically.

Thank you for the comment and suggestion. We revised this sentence into: “It can be seen from its cross-sectional view that the thickness of the oxide layer shows a trend that first increases then decreases with the Zr2Al4C5 content increased, the composite ceramic with 30 vol.% Zr2Al4C5 showed the thinnest oxide layer.”

10. In Figure 11, brightness and/or contrast must be corrected, because the images look almost black giving no chance to distinguish the element’s mapping.

   Thank you for the comment and suggestion. We revised the figure 11.

11. In Figure 12, EDX spectra are too small for perception.

Thank you for the comment and suggestion. We revised the figure 12.

We have rearranged text body and marked all revisions in red. Hopefully the revision will be satisfied.

Sincerely,

Reviewer 4 Report

The authors of the manuscript “Effect of Zr2Al4C5 content on the mechanical properties and oxidation behavior of ZrB2-SiC-Zr2Al4C5 ceramics” have done a good job writing the effect of including a Zr2Al4C5 in a ZrB2-SiC matrix. Even so, there are some points in the manuscript that must be corrected before this work will be published in materials.

1) The section “2.1 Preparation” is necessary to rewrite in a detailed way the preparation process of the mixture of raw materials. Thus, the paragraph in lines 111-116 is practically a copy of the text presented in reference [16]. From this paragraph, it is not clear which concentration of ZrB2 was in the initial mixture. In addition, it is necessary to maintain a similar style for all the components used, that is, indicate only vol% or wt% for each of the components.

2) In the section "2.1 Preparation" it is necessary to add a detailed description of the preparation process of the mixture to be sintered. For example, describe the ball milling process in detail indicating the equipment, and the parameters used such as: size of balls, material of the balls, dry or wet medium, type of medium, ratio of the medium and balls with respect to the mass of the dry powders. In addition, the post process used after ball milling must be indicated: drying (type and regimes) and sieving. This information is not available in reference [16].

3) In the same way, in the section “2.1 Preparation” is necessary to add a detailed description of the sintering process (equipment, sintering regimes: heating rates, pressure, sample diameter, sintering time, medium, etc.), to be able to repeat the experiment in other laboratories.

4) In the section “2.2. Characterization and Measurement” to describe the process of measuring Young's modulus. The reference [27] is not an open resource.

5) In the same way, in section “2.2. Characterization and Measurement" should describe the processes to determine the resistance to oxidation, heating processes, etc.

6) Line 157 – there is an error in the open porosity value.

7) Figure 1 – use another drawing to represent the temperature line.

8) Lines 186-187. Indicate these phases in at least one of the images in Fig. 3.

9) Lines 215-216: “It can be seen the ZrB2 grain size of ZAC0 sample is large, as shown in Fig.4(a), and its fracture mode is dominated by trans-granular fracture”. In these figures it is impossible to distinguish any of the phases studied.

10) Images 8, 9 and 10 are very small and cannot be seen correctly, please enlarge them. Especially fig.10 since none of the spectra can be seen.

11) I recommend adding an image that shows a relationship between the thickness of the layer vs. the concentration of Zr2Al4C5.

Author Response

Dear reviewer,

We greatly appreciate your valuable comments and great suggestions. After careful discussion, we would like to submit our revision with the detailed response to the questions as followings.

1. The section “2.1 Preparation” is necessary to rewrite in a detailed way the preparation process of the mixture of raw materials. Thus, the paragraph in lines 111-116 is practically a copy of the text presented in reference [16]. From this paragraph, it is not clear which concentration of ZrB2 was in the initial mixture. In addition, it is necessary to maintain a similar style for all the components used, that is, indicate only vol% or wt% for each of the components.

Thank you for the comment. We shortened the sentences. We revised the sentence into: “The designed component of composite ceramics is constant 20 vol% SiC and a total 80 vol% of ZrB2 plus Zr2Al4C5. The Zr2Al4C5 grains are synthesized by in situ reaction of Zr, Al and graphite powders (molar ratio is 2:6.2:4.8), and the Si powder (4 wt%) is added as a sintering additive for stabilizing the structure of Zr2Al4C5 compounds, and can replace Al solid solution into Zr2Al4C5 to stabilize the crystal lattice [28].”

The Si powder (4 wt%) was added as a sintering additive for stabilizing the structure of Zr2Al4C5 compounds, 4 wt% of the total weight of Zr, Al and C powders. Therefore we use the wt.%.

2. In the section "2.1 Preparation" it is necessary to add a detailed description of the preparation process of the mixture to be sintered. For example, describe the ball milling process in detail indicating the equipment, and the parameters used such as: size of balls, material of the balls, dry or wet medium, type of medium, ratio of the medium and balls with respect to the mass of the dry powders. In addition, the post process used after ball milling must be indicated: drying (type and regimes) and sieving. This information is not available in reference [16].

Thank you for the comment and suggestion. We revised the sentence into: “ ZrB2, SiC, Zr, Al, C and Si are weighed according to the component design, and they are mixed evenly by planetary ball mill (Vario-Planetary Mill, Fritsch Pulverisette 4, Germany) under an argon atmosphere at the speed of 400 r/min, using the WC grinding media, with the ball-to-powder weight ratio was 6:1. The mixed powder slurry was dried by rotary evaporator, and then passed through a 60 mesh sieve.”

3. In the same way, in the section “2.1 Preparation” is necessary to add a detailed description of the sintering process (equipment, sintering regimes: heating rates, pressure, sample diameter, sintering time, medium, etc.), to be able to repeat the experiment in other laboratories.

Thank you for the comment and suggestion. We revised the sentence into: “The as-synthesized powders were then loaded into a graphite die with a diameter of 15 mm, and sintered using a SPS system (model-1050, Sumitomo Coal Mining Co. Ltd., Tokyo). The temperature was measured by an optical pyrometer focused on the surface of the graphite die. The samples were heated to 600 oC at a rate of 300 oC/min, then with an average heating rate of 100 oC/min maintained upto 1800 oC and then the temperature is held constant for 3 min. The sample was cooled naturally after the sintering period was over. A uniaxial pressure of 20 MPa and a vacuum atmosphere were applied from the start to the end of the sintering cycle.”

4. In the section “2.2. Characterization and Measurement” to describe the process of measuring Young's modulus. The reference [27] is not an open resource.

Thank you for the comment and suggestion. We revised the sentence into: “The Young’s modulus (E) of the composites was determined using an ultrasonic equipment (Panametrics 5072PR) with a fundamental frequency of 20 MHz,”

5. In the same way, in section “2.2. Characterization and Measurement" should describe the processes to determine the resistance to oxidation, heating processes, etc.

Thank you for the comment and suggestion. We revised the sentence into: “A cuboid test block with a size of 4 cm×4 cm×3 cm was prepared by the same method in methods (1), and its surface area was expressed by S (unit: cm2). After weighing the initial mass M0 (unit: mg) of the test block using a high-precision balance (model: BS210S, Germany). Before oxidation, they were cleaned in an ultrasonic bath with alcohol. After dried, specimens were placed on a zirconia support with ridges to minimize the contact area. And oxidation tests were conducted in an electric furnace (model Nabertherm LHT04, Germany). Specimens were heated at 5 oC/min to different temperatures and held for 30 min in static air. And the mass M1 (unit: mg) of the test block is weighed again after calcination. And by formula (1), the oxidative weight gain ΔM per unit area of the test block is calculated.”

6. Line 157 – there is an error in the open porosity value.

Thank you for the comments and suggestions. We revised “0.29%” into “0.22%” at Lines 157.

7. Figure 1 – use another drawing to represent the temperature line.

Thank you for the comment and suggestion. We revised the figure 1 as Displacement vs. Temperature.

8. Lines 186-187. Indicate these phases in at least one of the images in Fig. 3.

Thank you for the comment and suggestion. We revised the figure 3.

9. Lines 215-216: “It can be seen the ZrB2 grain size of ZAC0 sample is large, as shown in Fig.4(a), and its fracture mode is dominated by trans-granular fracture”. In these figures it is impossible to distinguish any of the phases studied.

Thank you for the comment and suggestion. We agree with the views of the reviewers. We revised the sentence into: “It can be seen the ZrB2 grain size of ZAC0 sample is large, as shown in Fig.4(a), and the fracture of the complex ceramic is relatively flat.”

10. Images 8, 9 and 10 are very small and cannot be seen correctly, please enlarge them. Especially fig.10 since none of the spectra can be seen.

   Thank you for the comment and suggestion. We revised the figure 8, 9 and 10.

11. I recommend adding an image that shows a relationship between the thickness of the layer vs. the concentration of Zr2Al4C5.

Thank you for the comment and suggestion. We added the Figure 5 of the thickness of the layer vs. the concentration of Zr2Al4C5. We revised the sentence into: “Fig.6 shows the change of oxidative weight gain and oxide layer thickness of samples under different Zr2Al4C5 contents. It can be seen from the Fig.6 that with the increase of oxidation temperature, the oxidative weight gain and oxide layer thickness of all samples shows a gradual increase trend, especially when the temperature is higher than 1200 °C, the oxidative weight gain and oxide layer thickness increase trend is more obvious. With the increase of Zr2Al4C5 content, the oxidative weight gain and oxide layer thickness showed a trend of first increasing, then decreasing and then increasing, and when the dosage was 30 vol.%, the oxidative weight gain was the smallest, which was 16.2 mg/cm2 at 1500 oC.”

We have rearranged text body and marked all revisions in red. Hopefully the revision will be satisfied.

Sincerely,

Round 2

Reviewer 3 Report

Dear Authors,

Thank you for the corrections you provided in the manuscript.

However, Figures 10 and 12 are still far from being suitable for the readers. In the EDX spectra, the font is too small except to the elements' symbols. Please be respective to your potential audience and prepare the figures so that they were readable. In case the figures contain any excessive information just remove it.

A grammar check by a native English speaker is recommened.

Author Response

Dear reviewer,

We greatly appreciate your valuable comments and great suggestions. After careful discussion, we would like to submit our revision with the detailed response to the questions as followings.

However, Figures 10 and 12 are still far from being suitable for the readers. In the EDX spectra, the font is too small except to the elements' symbols. Please be respective to your potential audience and prepare the figures so that they were readable. In case the figures contain any excessive information just remove it.

Thank you for the comments and suggestions. We revised the figure 10 and 12, and especially in the EDX spectra.

We have rearranged text body and marked all revisions in red. Hopefully the revision will be satisfied.

Sincerely,
